# *Post Mortem* Study on the Effects of Routine Handling and Manipulation of Laboratory Mice

**DOI:** 10.3390/ani12233234

**Published:** 2022-11-22

**Authors:** Charles-Antoine Assenmacher, Matthew Lanza, James C Tarrant, Kristin L Gardiner, Eric Blankemeyer, Enrico Radaelli

**Affiliations:** 1Comparative Pathology Core, School of Veterinary Medicine, University of Pennsylvania, Philadelphia, PA 19104, USA; 2Penn State Health Milton S Hershey Medical Center, Penn State College of Medicine, Hershey, PA 17033, USA; 3GSK, Collegeville, PA 19426, USA; 4University Laboratory Animal Resources, University of Pennsylvania, Philadelphia, PA 19104, USA; 5Department of Radiology, University of Pennsylvania, Philadelphia, PA 19104, USA

**Keywords:** laboratory mouse, handling, restraint, manipulation, osteoarticular trauma, histopathology

## Abstract

**Simple Summary:**

Any routine manipulation and handling of mice for husbandry and/or experimental needs can cause physical harm. Besides being an ethical concern, trauma from manipulation and handling is a possible confounding effect on research outcomes. Specific pathologic effects of routine manipulations and handling are poorly documented and largely ignored by investigators. Our study provides a comprehensive *post mortem* overview of the main lesions associated with common manipulations of laboratory mice (i.e., restraint, blood drawing, and intraperitoneal injections), with an emphasis on traumatic osteoarticular changes. A total of 1000 mice were included, with 864 animals being heavily manipulated and 136 being handled for routine husbandry procedures only. Traumatic osteoarticular lesions were found in 61 heavily manipulated mice, and in a single unmanipulated mouse, showing a highly significant association between the presence of traumatic osteoarticular lesions and heavy handling of mice. We discuss the need for intentional training of research personnel on appropriate mouse handling and restraint techniques, which could help reduce the lesion frequency and the impact on animal wellbeing as well as study reproducibility.

**Abstract:**

Routine handling and manipulation of laboratory mice are integral components of most preclinical studies. Any type of handling and manipulation may cause stress and result in physical harm to mice, potentially leading to unintended consequences of experimental outcomes. Nevertheless, the pathological effects of these interventions are poorly documented and assumed to have a negligible effect on experimental variables. In that context, we provide a comprehensive *post mortem* overview of the main pathological changes associated with routine interventions (i.e., restraint, blood drawing, and intraperitoneal injections) of laboratory mice with an emphasis on presumed traumatic osteoarticular lesions. A total of 1000 mice from various studies were included, with 864 animals being heavily manipulated and 136 being handled for routine husbandry procedures only. The most common lesions observed were associated with blood collection or intraperitoneal injections, as well as a series of traumatic osteoarticular lesions likely resulting from restraint. Osteoarticular lesions were found in 62 animals (61 heavily manipulated; 1 unmanipulated) with rib fractures and avulsion of the dens of the axis being over-represented. Histopathology and micro-CT confirmed the traumatic nature of the rib fractures. While these lesions might be unavoidable if mice are manipulated according to the current standards, intentional training of research personnel on appropriate mouse handling and restraint techniques could help reduce their frequency and the impact on animal wellbeing as well as study reproducibility.

## 1. Introduction

Mice have been used in biomedical research for centuries, with the first reports of scientific experiments using mice dating to the 17th century [1]. Since then, the usage of mice dramatically increased, both in absolute numbers and in the type of experiments. A recent report roughly estimated the number of mice and rats used between 2017 and 2018 to be over 111 million in the USA alone [2]. This is however most likely an underestimation given the lack of required reporting of mice and rats according to the Animal Welfare Act (AWA) [2]. Manipulation of mice is an integral part of most research and ranges from gentle handling for bedding and cage changes, to varying degrees of traumatic manipulations including various restraint methods, blood collections, intravenous or intraperitoneal injections, surgical procedures, etc. Any type of manipulation has the potential to cause stress and result in physical harm to mice [3,4]. While mice are known to be stoic and only show signs of distress when the damage is severe and potentially irreversible, the resulting effects of traumatic manipulations remain an ethical concern and may have a significant confounding effect on research outcomes [5]. A large body of literature exists on the effects of handling and experimental procedures on laboratory animals’ well-being and stress level [4,6,7,8]. However, the specific pathologic effects of these manipulations are far less documented and largely ignored by investigators. Hence, the impact of routine animal manipulations is likely to be overlooked or considered negligible in most experimental contexts.

The present study provides a comprehensive *post mortem* overview of the main pathologic changes associated with routine handling and restraint techniques for experimental interventions in laboratory mice with an emphasis on traumatic osteoarticular lesions. The application of a rigorous and comprehensive protocol for *post mortem* analysis of experimental mice across different studies (which includes a thorough macroscopic examination and histopathological evaluation of all major organs) has been critical to defining many of these manipulation-induced lesions. While some lesions are expected based on the type of manipulations (e.g., blood collection procedures or intraperitoneal injections), this study also reveals unique traumatic osteoarticular lesions that are unexpectedly associated with standard handling/restraining techniques.

## 2. Materials and Methods

### 2.1. Animals

The mice considered in this retrospective study were selected from the complete pathological assessment records of the Penn Vet Comparative Pathology Core at the University of Pennsylvania. The same procedure for euthanasia, necropsy, trimming, and histopathological examination was applied to all mice included in this study. Euthanasia was achieved by progressive CO_2_ inhalation, and exsanguination via cardiac blood collection served as a secondary method of euthanasia. Complete necropsy with macroscopic *post mortem* examination was then performed and the following organs/tissues were sampled and fixed in 10% neutral buffered formalin (NBF) for histopathological examination: head and brain, spine with spinal cord (including cervical, thoracic, and lumbar segments), skin from the dorsal region, pinnae, quadriceps femoris, sternum, salivary glands, larynx and trachea, esophagus, lungs, heart, gastrointestinal tract, mesenteric ligament, pancreas, liver and gall bladder, gonads and reproductive tract, kidneys, urinary bladder, and any organ/tissue showing macroscopic lesion/s. The weights of liver, spleen, right and left kidneys, and heart were recorded before fixation. Formalin-fixed tissue samples were trimmed according to the RITA guidelines (including minimal modifications) [9,10,11] and then routinely processed for paraffin embedding, sectioning and H&E staining. Histopathological examination was performed by four board-certified veterinary pathologists (CAA, ML, JCT, and ER).

### 2.2. Review of Reports

To assess the prevalence of manipulation-induced lesions in mice, all reports resulting from the complete pathological assessment of mice from 2017 to 2022 were reviewed. All data regarding handling, type of experimental manipulation, strain, age, and sex were recorded. Mice were grouped into three age categories according to previous studies [12,13]: juvenile (<10 weeks old, 46 females and 53 males); young adult (10–26 weeks old, 461 females and 224 males); adult (>26 weeks old, 129 females and 87 males). For all mice, relevant lesions were extracted from the pathology narrative portion of the report and categorized as (1) osteoarticular trauma (including bone fracture, joint dislocation, or any combination of the two), (2) penetrating trauma to the ocular/periocular region, (3) penetrating trauma to the facial and maxillary regions, (4) penetrating trauma to the abdomen resulting in peritonitis, hemoabdomen and/or puncture of viscera.

Relevant information regarding husbandry, type of manipulation, and handling was collected from the investigator’s protocols and/or from the disclosure of experimental procedures provided by the investigators at the beginning of each individual project.

### 2.3. Micro-CT

A U-CT high-speed and high-resolution scan (MILabs; Houten, The Netherlands) using the mouse ultra-focus mode was used for the imaging of the thoracic osteoarticular lesions.

### 2.4. Statistical Analysis

Statistical analyses were performed using GraphPad Prism 9.4 software. Fisher’s exact test was used to analyze variations in the frequency of lesions across different mouse groups. Chi-square test for trend was used to compare the frequency of traumatic lesion in manipulated mice across different age categories.

## 3. Results

### 3.1. Demographics of the Study Population

A total of 1000 mice were necropsied between 2017 and 2022, 636 females and 364 males. A number of strains were represented, including NOD.Cg-*Prkdc^scid^ Il2rg^tm1Wjl^*/SzJ (NSG^TM^; referenced as NSG for the rest of the manuscript) (*n* = 574), C57BL/6J and genetically engineered mice on a C57BL/6J background (*n* = 370), CD1 (*n* = 20), DBA/2 (*n* = 12), Swiss outbred (*n* = 12), C3HF1 (*n* = 6) and BALB/cJ (*n* = 6). The records indicated that 864 mice were subjected to one or more of the following experimental manipulations as part of the study protocol(s): intravenous (IV) (*n* = 661) or intraperitoneal (IP) (*n* = 509) injections, retro-orbital (RO) (*n* = 317) or facial/maxillary vein (cheek bleed (CB)) (*n* = 164) blood collection(s) or other manipulation (subcutaneous tumor cell injections, extensive behavioral testing, etc.). These manipulations were often repeated/combined multiple times throughout the course of the experiments as well as accompanied by forced physical restraint and therefore considered as heavy manipulations. A minor proportion (*n* = 136) were handled only for routine husbandry purposes (i.e., cage changes and cleaning).

### 3.2. Lesions Associated with Manual Handling/Restraining

Out of the 864 mice that were subjected to experimental manipulation other than for routine husbandry purposes, 61 exhibited at least one of the following traumatic bone and articular damages: transverse and comminuted fractures of ribs (*n* = 24) (Figure 1a–c), sternebrae (*n* = 5) and sacrum (*n* = 1); avulsion fractures with chondromucinous degeneration primarily affecting the 2nd cervical vertebra (axis) (*n* = 22) (Figure 2) and other vertebral processes (*n* = 5), and articular dislocation with secondary cartilaginous degeneration of chondrocostal (*n* = 3) and sacroiliac joints (*n* = 1).

In contrast, among the 136 mice that were not subjected to any experimental manipulation, only a single male genetically engineered mouse on a C57BL/6J background, exhibited traumatic bone damage [rib and sternebral fractures (*n* = 1)]. The association between heavy manipulation and the presence of osteoarticular lesions was statistically highly significant (Figure 3a) (*p* = 0.0018; Fisher’s exact tests). To better characterize the extent of the trauma, a correlative micro-CT study was performed on mice with macroscopically identifiable rib lesions. The micro-CT confirmed the presence of complete transverse rib fractures with callus formation (Figure 1c). Male sex was significantly more frequently associated with traumatic bone and articular lesions, despite the fact that similar experimental interventions were applied to both males and females (Figure 3b) (*p* = 0.0253; Fisher’s exact test). The effect of the genetic background was assessed in the NSG and C57BL/6J groups as enough mice from multiple studies were available for these strains only. Mice from these two strains underwent similar types of experimental interventions. In this context, the frequency of osteoarticular lesions was significantly higher in mice on a C57BL/6J background as compared to NSG mice (Figure 3c) (*p* = 0.0046; Fisher’s exact test). When mice were grouped and compared based on the type of experimental manipulation, no significant differences were seen between mice undergoing retro-orbital or facial/maxillary vein bleeding, and IV or IP injections (Figure 3d–g) (RO: *p* = 0.0671, CB: *p* = 0.7250, IV: *p* = 0.0708, IP: *p* = 0.4316; Fisher’s exact test). While an overall trend towards increased traumatic osteoarticular lesion frequency in older animals was observed, statistical significance was not reached, even after stratification based on strain and sex (Appendix A).

### 3.3. Blood Collection-Related Findings

As expected, numerous mice undergoing blood collection as part of the experimental protocol exhibited changes attributable to the specific technique applied. These changes were categorized as (1) penetrating trauma to the ocular/periocular region (for retro-orbital bleeding) and (2) penetrating trauma to the facial and maxillary regions (for facial vein bleeding). The former was seen in 133 mice out of 317 with confirmed RO blood collection and ranged from mild inflammatory cell infiltrates in the limbal conjunctiva and episclera, to extensive necrotizing and suppurative retro-orbital cellulitis with destruction of the Harderian gland, or well-defined retro-bulbar abscesses. Penetration of the sclera with secondary panophthalmitis and *phthisis bulbi* was noted on occasion (Figure 4). Foreign material, such as bedding/plant material or hair shafts, as well as bacterial or fungal organisms were frequently noted amidst the inflammatory infiltrates (Figure 4). Exposure keratitis, presumably due to the excessive retro-orbital pressure and anterior displacement of the globe was also occasionally seen.

Similarly, 39 out of 164 mice that underwent blood collection by puncture of the maxillary, or another branch of the facial vein, exhibited pathologic changes in the region. The changes ranged from minor venous non-occlusive mural thrombi (Appendix A) to extensive vascular damage with perivascular hemorrhage, necrosis and granulation tissue (Figure 5).

In some mice, the necrosis extended to the adjacent tissues, with widespread necrosis of the parotid salivary or extraorbital lacrimal glands (*n* = 3) (Appendix A). In most severe cases, the puncture extended into the deeper tissues, with occasional penetration of the calvarium bones and linear tracts into the underlying cerebral hemisphere (*n* = 13) (Figure 6). These linear tracts were associated with one or more of the following: regional loss of cortical neurons, glial scar formation, the introduction of foreign material (most frequently hair shafts), accumulation of hemosiderophages, and/or subdural hemorrhage. In a single mouse, this blood collection method was suspected to be the cause of premature and unanticipated death (Appendix A).

### 3.4. Peritoneal Injection-Related Findings

Out of the 509 mice receiving IP injections, 92 exhibited one or more gross and/or histopathological signs consistent with penetrating abdominal trauma. These included variable degrees of peritonitis (*n* = 82) or hemoabdomen (*n* = 21), puncture of visceral organs (*n* = 16), and necrosis and inflammation of the mammary fat pad (*n* = 6). In the cases of peritonitis, the inflammation was frequently associated with the introduction of foreign material (i.e., hair shafts) and bacteria, or with the injection of lipid-rich vehicles, resulting in a granulomatous inflammatory process with intralesional lipid droplets. The puncture of visceral organs, including the gastrointestinal or reproductive tracts, the kidneys or the liver commonly resulted in a localized inflammatory reaction with linear tracts of necrosis and/or granulation tissue in the parenchyma, and introduction of foreign material (i.e., hair shafts or refractile material of unknown origin) (Figure 7 and Figure 8). In two mice, an IP injection leading to hemoabdomen was the suspected cause of unanticipated death (Appendix A).

## 4. Discussion

Mice have been used in biomedical research for centuries [1]. Because of their widespread use in diverse experimental settings, mice are exposed to various levels of handling and restraint. In this study, we provide a comprehensive description of the nature and frequency of lesions due to common laboratory procedures such as restraint, blood collections, and intraperitoneal injections.

This study presents obvious limitations due to the retrospective nature of the data. In particular, the nonuniform dataset, including undetailed experimental intervention history for some of the animals, prevented a more granular assessment of the relative contribution of different variables to the development of the reported lesions. Nonetheless, we believe the sheer number of animals included in this study and the statistically significant correlation between forced physical restraint and specific types of traumatic osteoarticular lesions support our hypothesis of handling-induced pathogenesis.

A multitude of methods have been described for handling and restraint of mice. Those practices are essential for the safety of the animals as well as the handler [14]. Nevertheless, although the negative effects of certain restraining techniques on animal well-being and overall health have been known and studied for decades [15], limited literature exists on the pathological consequences of routine handling of mice. Importantly, there are no studies specifically reporting or investigating the potentially traumatic effects of those manipulations. In this regard, for the first time, our study clearly delineates a causative association between the occurrence of a series of traumatic osteoarticular lesions and restraining.

The distribution of these traumatic lesions, as well as the highly significant association with repeated restraint, supports the notion that forced handling is the critical determining factor in their development. Male mice and mice on a C57BL/6J background were significantly more likely to present osteoarticular lesions. The exact reason for these correlations remains unclear. Mice on a C57BL/6 background are not classically considered as being more aggressive [16], which reduces the possibility of difficult handling and increased use of force as a cause for the osteoarticular lesions. This strain is however known to have a lower bone density when compared to other inbred strains [17], which may predispose them to increased bone fragility. On the contrary, NSG mice have not been reported as having decreased bone density, even under experimental conditions that require whole body irradiation [18]. In addition, while a reduction in bone density has been described in NOD mice as a complication of diabetes mellitus [19], NSG mice are known to be resistant to the development of type 1 diabetes mellitus due to the inability to mount a T cell response and are therefore not expected to develop reduced bone density as an effect of the disease [20].

Adissu et al. described sternal lesions in C57BL/6N mice in the context of an extensive phenotyping program with frequent handling and restraint of animals. The lesions consisted of dislocation of the 4th sternebra and were seen with a higher prevalence in males. They hypothesized a possible traumatic pathogenesis caused by the frequent neck scruffing and subsequent increased constraint on the sternum [21]. Interestingly, sternal lesions consistent with those described by Adissu and coauthors, were seen in 5 mice in our study and all were in mice on a C57BL/6J background except one in a CD1 mouse.

Blood collection is a crucial part of most scientific investigations and can be performed in several ways, all of which require adequate and specific training [22]. Techniques most commonly used in our database included orbital venous sinus and facial/maxillary vein puncture. Clinical consequences frequently reported for these procedures include bleeding from the ear canal, temporary signs of discomfort (i.e., inactivity, hunched posture) or even unexpected death for facial vein puncture, and periocular discharge, en- or exophthalmos with or without corneal and intraocular lesions for retro-orbital bleeding [22,23,24,25,26]. Some studies report a complication rate as low as 1–2% [27], while others are more realistic and report variable complication rates depending on the operator’s experience [25]. Severe complications might remain undetected due to subtle external signs which may be missed by laboratory technicians [28]. While the vast majority of histopathological changes seen in the animals included in our study were minor and expected, a non-negligible proportion of mice exhibited severe signs which resulted in death or can reasonably be expected to have had an effect on the animal’s well-being. Different blood collection techniques are known to have advantages and disadvantages and conflicting data exist regarding the effects of individual techniques on the animal’s well-being [29,30]. Retro-orbital sinus puncture, for example, was reported as causing less stress and tissue damage in mice compared to other blood collection techniques, with or without anesthesia [29], while this technique was associated with increased plasma and fecal corticosterone levels and increased behaviors associated with discomfort in another study [31]. The lack of a definitive scientific consensus on this matter resulted in substantial discrepancies in terms of regulations in force across different countries. For example, per Canadian regulations, retro-orbital sinus bleeding is allowed only as terminal procedure, while the same procedure can be applied as a survival intervention in US [32,33]. This example highlights the need for more comprehensive studies with systematic assessment of each technique in diverse experimental contexts to guide investigators towards the most appropriate and less harmful procedure.

Peritoneal injections are an integral part of research and are performed for various reasons, whether for injection of experimental drugs, for injections of medications such as tamoxifen, busulfan, anesthetics, or for orthotopic implantation of neoplastic cells. When performed appropriately, these manipulations typically result in minimal changes. However, our data showed the development of significant pathological changes of the peritoneal cavity and/or abdominal wall in almost 20% of the mice receiving IP injection(s). In some cases, such as with the injection of lipid-rich vehicles, the changes were anticipated. In others, they are suspected to be due to poor technique with the introduction of foreign material or bacteria, and puncture of visceral organs.

Although we were not able to detect a statistically significant correlation between the different blood collection techniques, the IV and IP injections, and the presence of traumatic osteoarticular lesions, the vast majority of the manipulated animals included in this study underwent either all or a combination of these experimental interventions. The relative contribution of each individual manipulation to the overall spectrum of osteoarticular traumatic changes is therefore difficult to ascertain.

While most of the pathological changes associated with blood collection and IP injections have been reported in the literature and studied extensively in regard to animal welfare, the osteoarticular lesions we report, and their potential effects on research outcomes are not known, and likely not taken into consideration by investigators. The consequences of pain and analgesia on several physiologic and pathologic conditions has been extensively investigated, with effects ranging from suppression of specific immune functions [34,35], modulation of cancer biology [36,37,38] or significant alteration of normal behavior [39].

The pain generated by bone fractures is well-studied in the context of fracture repair research [40]. Importantly, undetected accidental fractures have the potential to perturb a wide range of physiological/experimental parameters including the measurement of serum alkaline phosphatase [41], cytokine profiles [42,43,44], acute phase proteins [45], or leukocytes/immune cells composition in the peripheral blood and bone marrow [46,47]. The cardiac function can also be compromised in the case of sternal lesions impinging on the overlying thoracic organs [21]. In addition, while not clinically manifest, osteoarticular trauma can interfere with a series of behavioral tests, cognitive tasks, and neuromuscular functions [48]. In this regard, the fractures/dislocations noted in our study may induce significant and unexpected confounding factors and it is critical to raise the awareness within the biomedical research community of the possibility to inadvertently induce these traumatic osteoarticular lesions when handling mice.

A recent survey of 185 laboratory animal veterinarians identified that the training of researchers performing procedures on animals was the number one area that would benefit from continued evaluation in order to increase mouse welfare in research as well as experimental reproducibility [49]. Intentional training of investigators is crucial to master the skills needed to properly execute all these routine procedures and has been shown to yield excellent results when performed appropriately in the context of orbital sinus puncture for example [26]. Therefore, we strongly believe mandating structured training program which encompasses basic animal anatomy and physiology, thorough supervised hands-on training, competency assessment as well as recurring reassessments is crucial to improve the investigators’ technique and reduce the risk of adverse effects on these stoic but small and fragile animals. Using templates for the assessment of proficiency and competency of individual investigators, similar to what Clifford and coauthors suggested [50] may help in that regard. Additionally, developing the investigators’ ability to detect decreases in animal welfare in response to post-procedural pain (i.e., Grimace scales) [51] should be a crucial part of the training and may be helped by new approaches which reduce inter-observer variability such as algorithm-measured voluntary wheel running for example [52]. Finally, refining the current standards for mouse handling has been shown to reduce the level of stress associated with each human-mouse interaction [6,53,54], given the clear association we show between manipulation and the presence of traumatic osteoarticular lesions, a revision of experimental protocols aiming at reducing the number of certain manipulations will likely be beneficial for both animals and research.

## 5. Conclusions

In conclusion, our results establish a very compelling causative link between specific routine manipulations and clinically significant lesions in laboratory mice. Therefore, we believe that the outcomes of our study are important to inform the refinement of best practices for routine mouse handling/restraining, blood collection, and IP injection with the ultimate goals of minimizing animal discomfort/pain and improving experimental reproducibility.

## Figures and Tables

**Figure 1 animals-12-03234-f001:**
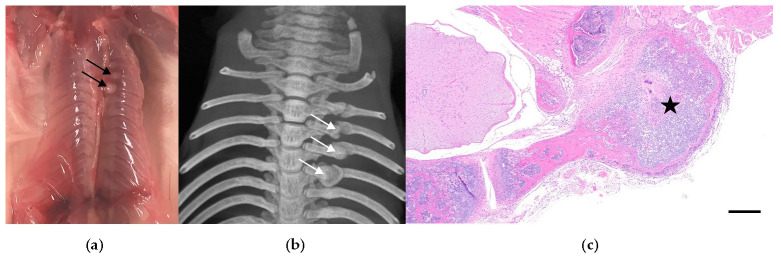
Example of rib fracture in a heavily manipulated mouse. (**a**) Macroscopic image showing multiple rib fractures (black arrows). (**b**) Micro-CT scan of the thorax confirming the multiple rib fractures (white arrows). (**c**) H&E photomicrograph of one of the affected ribs showing the fracture site with osteocartilaginous callus formation (star). Scale bar, 300 μm.

**Figure 2 animals-12-03234-f002:**
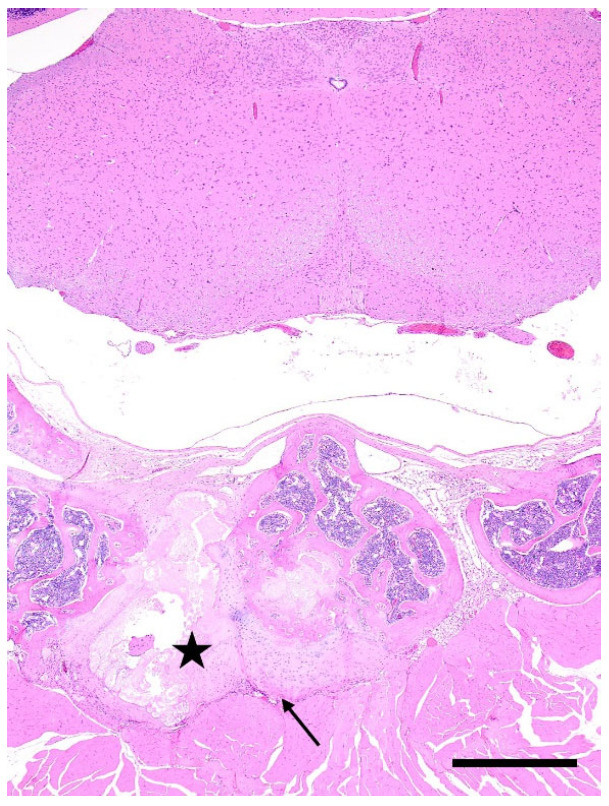
Example of avulsion fracture of the axis (2nd cervical vertebra). Photomicrograph of the atlanto-occipital and atlanto-axial joints. The bone of the axis is regionally fractured with chondromucinous degeneration (star), and cartilaginous callus formation (black arrow). H&E stain. Scale bar, 500 μm.

**Figure 3 animals-12-03234-f003:**
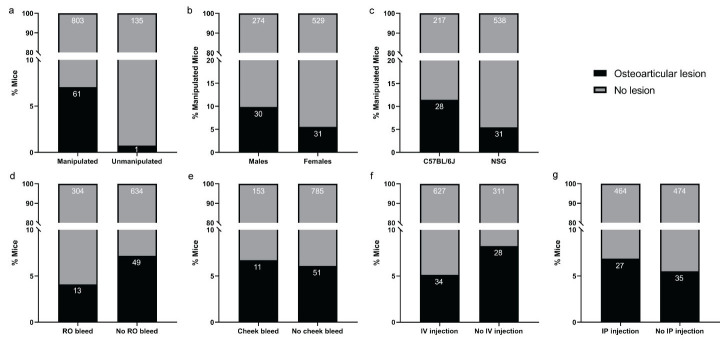
Statistical comparisons of the frequency of traumatic osteoarticular lesions. Statistical comparisons of the frequency of traumatic osteoarticular lesion between: (**a**) manipulated and unmanipulated mice (*p* = 0.0018 **), (**b**) male and female manipulated mice (*p* = 0.0253 *), (**c**) NSG and C57BL/6J manipulated mice (*p* = 0.0046 **), (**d**) mice undergoing retro-orbital bleeding or not (*p* = 0.0671 ns), (**e**) mice undergoing cheek bleeding or not (*p* = 0.7250 ns), (**f**) mice undergoing tail vein injections or not (*p* = 0.0708 ns), and (**g**) mice undergoing intraperitoneal injection(s) or not (*p* = 0.4316 ns). ns *p* > 0.05; * *p* ≤ 0.05; ** *p* ≤ 0.01 by Fisher’s exact test. RO, retro-orbital; IV, intravenous; IP, intraperitoneal; ns, no significance.

**Figure 4 animals-12-03234-f004:**
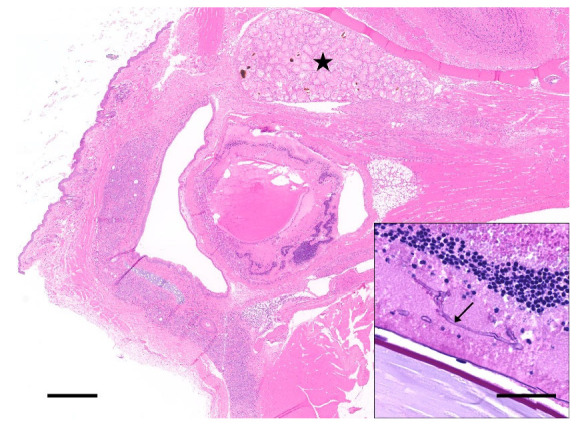
Severe ocular and periocular lesions after retro-orbital bleeding. Example of a severe case of ocular and periocular damage associated with retro-orbital bleeding in a mouse. The eye is small and shrunken (*phthisis bulbi*) with marked intraocular inflammation and intralesional fungal hyphae (black arrow, inset). The periocular tissues are severely inflamed and the Harderian gland is diffusely necrotic (black star). Left scale bar, 500 μm; Inset scale bar, 50 μm. Section in the inset is stained with a Periodic acid-Schiff (PAS) stain.

**Figure 5 animals-12-03234-f005:**
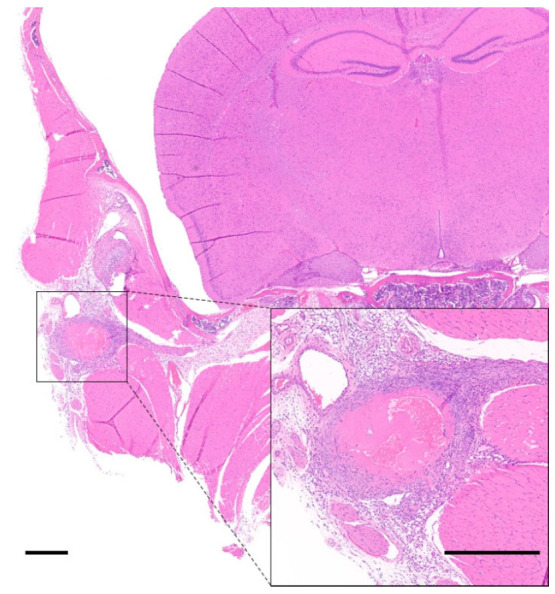
Lesion associated with facial vein puncture. Example of occlusive venous thrombus in the facial vein with abundant perivascular inflammation and granulation tissue. H&E stain. Scale bars, 500 μm.

**Figure 6 animals-12-03234-f006:**
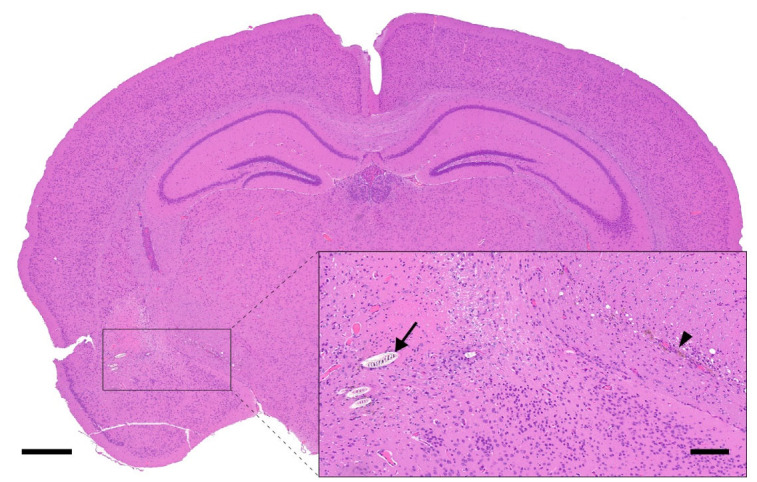
Deeper lesions associated with facial vein puncture. Photomicrograph of the brain of a mouse undergoing facial vein puncture with accidental penetration of the underlying cerebrum. The lesion in this mouse is associated with the introduction of hair shafts (black arrow, inset), mixed gliosis and minimal chronic hemorrhage (black arrowhead, inset). H&E stain. Left scale bar, 500 μm; Inset scale bar, 100 μm.

**Figure 7 animals-12-03234-f007:**
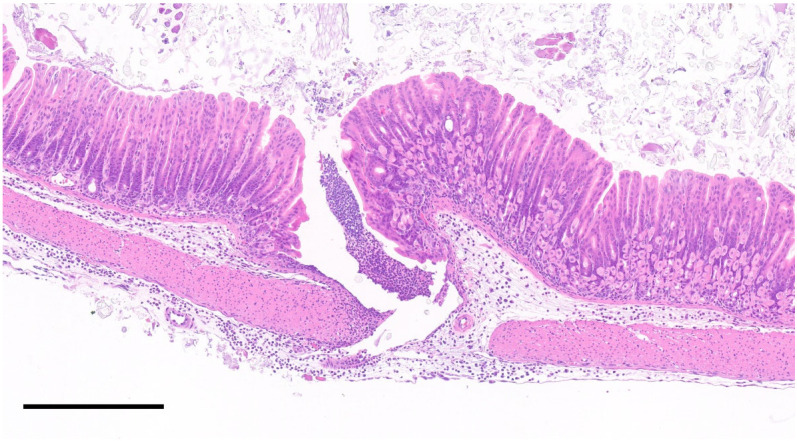
Gastric wall puncture associated with IP injections. Photomicrograph of the gastric wall of a mouse receiving repeated IP injections. The gastric wall has a focal transmural defect filled with a coagulum of neutrophils. Similar inflammatory infiltrates, along with edema expand the submucosa and the serosa. H&E stain. Scale bar, 300 μm.

**Figure 8 animals-12-03234-f008:**
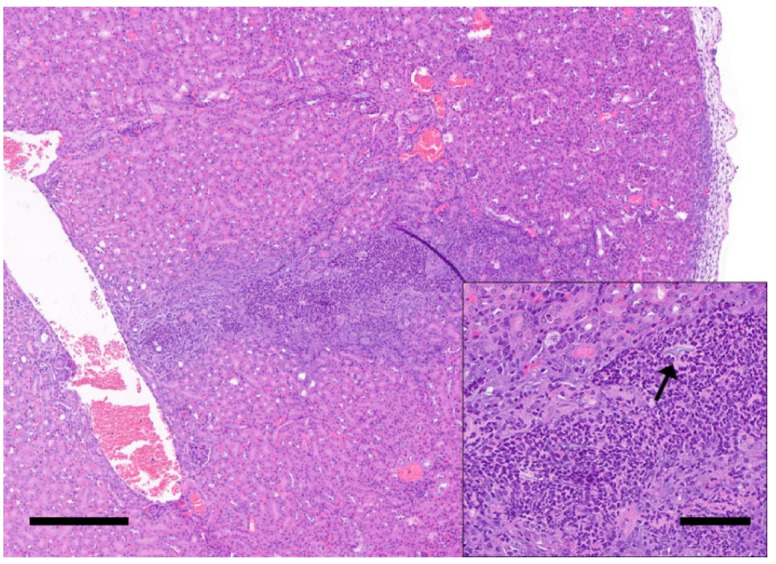
Renal puncture associated with IP injections. Photomicrograph of the caudal pole of the kidney of a mouse receiving repeated IP injections. The renal parenchyma is regionally distorted by a linear tract of neutrophilic inflammatory cell infiltrates with occasional intralesional foreign material (black arrow, inset). The tubules adjacent to the linear tract exhibit marked regenerative changes (i.e., tubular basophilia) and the overlying renal capsule is thickened by inflammatory cells and connective tissue. H&E stain. Left scale bar, 300 μm; Inset scale bar, 100 μm.

## Data Availability

Not applicable.

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
