# Peer review of "Post Mortem Study on the Effects of Routine Handling and Manipulation of Laboratory Mice"

_animals, 2022, doi:10.3390/ani12233234_

Round 1
Reviewer 1 Report
The paper describes the results of postmortem analysis of 1,000 experimental mice having undergone one or more (combinations of) manipulations in their lifetime. The authors report their results in a comprehensive manner and acknowledge the limitations of the study. The results are of interest to both researchers, educators, animal facility personnel and colleagues involved in IACUC/ Animal Welfare Body/ AWERB.
I take this opportunity to share some thoughts and make suggestions to the submitted manuscript.
Materials and Methods:
- Animals: add a figure with age distribution by sex. Age affects susceptibility to certain types of trauma/ lesions.
- Line 87 - abbreviation NBF not explained. General remark: introduce an abbreviation at first use unless one can be absolutely sure that the readership of the journal is expected to know e.g. DNA.
- Line 127 - Abbreviation NSGTM is introduced while in the rest of the manuscript this strain is referred to as NSG.
Results:
- The authors appear cautious in their handling and analysis of the data, which is to be commended. However, the paper would benefit if the data would allow for an analysis for example by sex and age group.
- Lines 163-164 and 167-168 - it is stated that male mice and B6 were significantly more frequently associated with traumatic bone and articular lesions. Have the authors ascertained themselves that the different types of manipulations were equally distributed across the sexes and across the genetic backgrounds? There is no mentioning of that in the manuscript while this information is crucial to verify the reported result.
- Lines 228 - 231 - 92 animals exhibited gross and histopathological signs. The breakdown amount to 125 cases. Add one sentence to indicate which ones were seen together in how many cases. If it is random, suggestion to state that in one sentence.
- Lines 286 - 289 - consider rephrasing to emphasise that a difference between NOD and NSG mice is that the latter do not develop diabetes, which is a contributing factor to decreased bone density.
- Lines 307-310 - this reviewer encourages the authors to take their conclusion a step further and call for review of existing practices. For example in Europe guidelines state that blood collection through orbital plexus puncture should be considered a terminal procedure.
- Lines 346-353 - this reviewer encourages the authors to take their conclusions on training one step further and advocate the need for a training framework that covers the whole training sequence from basic training without animals, to training on animals under supervision until competence is achieved with reassessments at certain frequencies.
Reviewer 2 Report
In this article, the authors present the results of a large-scale study aimed at assessing the damage caused by multiple manipulations of mice, including handling, restraint, injections and blood sampling. These "simple" manipulations are commonly practiced in animal research but also in teaching.
The article nicely presents the context and the methodology used. The illustrations are extremely evocative and appropriate. However, a few details are missing, especially concerning the definition of "heavy manipulation". It is regrettable that the exploitation of the data did not allow to determine the importance of the number of repetitions in the effects and damages observed. This is probably due to the great heterogeneity of the data set or maybe to a deliberate choice not to attach too much importance to it. Could the authors clarify their views on this point? Of course a single mishandling, for example by a less trained person, may be sufficient to create an injury that is not visible but serious enough to affect the welfare of the animal. However, would it be possible to clarify what is meant by "heavy manipulation"?
Judging from the quantitative data presented, some injuries are rare while others appear to be more frequent, especially with certain strains of mice that have known and documented vulnerabilities. This clearly seems to be the case with C57Bl/6 mice, whose bone density is lower than that of other strains and makes them more at risk for bone lesions and trauma. In terms of training and education, this observation should lead to a reinforcement of the acquisition of in-depth knowledge of the specificities of each mouse strain.
Figure 3 shows some very puzzling findings. It appears that the percentage of animals with a bone lesion/trauma is lower (4.27%, based on the numbers shown) for animals undergoing retro-orbital puncture than for animals that did not (7,72%). The same observation applies to animals undergoing or not undergoing an IP injection (5.42% versus 9%). Is this an error? If not an error, what would be the explanation?
The low number of lesions observed in the tail region is surprising. Tail-handling of mice is (unfortunately) still extremely common in routine operations such as litter changing, despite the abundant litterature drawing attention to this point. Do the authors have specific data related to this type of handling? Some strains of mice (Scid) are so fragile in this respect that suspension by the tail can lead to a complete detachment!
Rib fractures or dislocations are particularly spectacular and sufficiently serious that one can be surprised that clinical translation or behavioral signs of these lesions are never observed or reported. It is difficult to believe that such lesions are painless and do not affect respiratory function. This may be due to the fact that mice are prey and can mask or conceal certain lesions/harm, but also to the lack of criteria and time devoted to observing the animals once the handling/restraint have been performed.
In addition to encouraging training and regular practice of a given procedure, the discussion should emphasize the fragility of the animals and the need to take this vulnerability into account, as well as the need to look for clinical signs or criteria to look for in the minutes/hours following the performance of a given act. That an animal such as a mouse can be fragile may seem very obvious at first. However, this characteristic is often underestimated during initial training and progress in this area is still necessary.
Improvement in handling/restraint involves the development of realistic and therefore fragile animal substitutes/simulators, but also the use of animal cadavers that are kept as close to living condition as possible, which can then be examined and autopsied after a restraint or an injection.
The search for clinical signs may involve the use of video cameras to observe the animals without the presence of a human being near the cage, and also by carrying out simple tests: for example, comparing the way the animal walks in a handling tunnel before and after an injection, when it is returned to the cage.
It is clear from the results that the more an animal is handled, the greater the risk that the animal will suffer injury and trauma. Therefore, limiting the number of times an animal is handled or restrained becomes an ethical requirement and should contribute to reducing the harm inflicted. This should lead to a revision of experimental procedures, including those for regulatory testing, so as to limit the repetition of injurious acts.
On a personal note, I would like to thank the authors for this work, which should be undertaken more often in every research institution, to assess the quality of the execution of procedures and also the concern of operators for the welfare of the animals. In conclusion, this article is one of a few that should be a landmark and inspire further work and, above all, a real reflection on the consideration of the intrinsic vulnerabilities of the animals that we handle on a daily basis. As already stated by many authors, it is not only a question of animal welfare but also of the quality of the data obtained.
Reviewer 3 Report
General comment
The manuscript reports data regarding lesions induced in laboratory mice through routine handling and manipulation of laboratory mice. The study has been carried out on a considerable number of mice sumbitted to a complete post mortem (gross and histological, in some cases Micro-CT) examination.
The topic is of extreme interest from both the scientific and ethical point of view.
The manuscript is well written and the illustrations are excellent in most of the cases.
I have few minor suggestions which can be taken into account by the Authors.
Specific (minor) comment
Title: the term “postmortem” is used in several formats (postmortem, post-mortem, post mortem) in scientific literature. I suggest to use the original latin format: “post mortem” in the title and throughout the manuscript.
Line 84: CO2 with 2 in superscript
Line 87: NBF > spell out before the acronym
Lines 191 and 199: phthisis bulbi > in italics
Comments on the figures
Fig 4 - Fungal hyphae are no visible in the inset. I suggest to change or delete it.
Fig 8 – The black arrow is missing in the inset. Foreign materiali is barely visible. I suggest to increase the magnification.
